# In Vitro Comparison of Biocompatibility of Calcium Silicate-Based Root Canal Sealers

**DOI:** 10.3390/ma12152411

**Published:** 2019-07-29

**Authors:** Ju Kyung Lee, Sunil Kim, Sukjoon Lee, Hyeon-Cheol Kim, Euiseong Kim

**Affiliations:** 1Department of Conservative Dentistry, School of Dentistry, Dental Research Institute, Pusan National University, Yangsan 50612, Korea; 2Microscope Center, Department of Conservative Dentistry and Oral Science Research Center, Yonsei University College of Dentistry, 50-1 Yonsei-Ro, Seodaemun-Gu, Seoul 03722, Korea; 3BK21 PLUS Project, Yonsei University College of Dentistry, 50-1 Yonsei-Ro, Seodaemun-Gu, Seoul 03722, Korea; 4Department of Electrical & Electronic Engineering, Yonsei University College of Engineering, 50 Yonsei-Ro, Seodaemun-Gu, Seoul 03722, Korea

**Keywords:** biocompatibility, root canal sealer, calcium silicate, epoxy resin, cell viability, inflammatory response

## Abstract

The aim of this study was to assess the effect of three calcium silicate-based sealers (EndoSeal MTA, Nano-ceramic Sealer, and Wellroot ST) and two epoxy resin-based sealers (AH-Plus, AD Seal) on various aspects, such as cell viability, inflammatory response, and osteogenic potential, of human periodontal ligament stem cells (hPDLSCs). AH-Plus showed the lowest cell viability on hPDLSCs in all time periods in fresh media. In set media, hPDLSCs showed no significant differences in cell viability among all the tested materials. Wellroot ST showed the highest level of cell adhesion and the morphology of attached cells. AH-plus presented a significantly higher expression of IL-6 and IL-8 than the other sealers. AD Seal and three calcium silicate sealers showed high expression of the mesenchymal stem cell markers. ALP mRNA expression showed a significant increase in time-dependent manner on all of three calcium silicate-based sealers, which do not seem to interfere with the differentiation of hPDLSCs into osteoblasts. Based on the results from this study, calcium silicate-based sealers appear to be more biocompatible and less cytotoxic than epoxy resin-based sealers. Meanwhile, further and long-term clinical follow-up studies are required.

## 1. Introduction

Three-dimensional obturation of the root canal system is an important step in root canal treatment [1]. In the traditional canal obturation techniques, gutta-percha has been used as the main canal filling material, and root canal sealer has been used as an auxiliary/complementary material [2,3]. However, the contemporary concept of canal obturation has been changed with advancement of root canal sealer and its simplified application method [3]. Recently introduced calcium silicate or bioceramic-based sealers are suggesting the single cone technique, which may permit more sealers than the conventional techniques, and this technique is changing the paradigm of root canal filling procedure [3,4]. 

Ideally, the contact between the root canal wall and the filling material is terminated at the physiologic apical foramen area [2]. However, the sealer is open extruded over the apex in practice. When the sealer overfills or extrudes to the periapical area, unreacted material may stimulate the immune system, which causes inflammatory reaction to the apical tissue [5]. This undesirable reaction may happen via the accessory canals or dentinal tubules [2]. Even if there is no extrusion, root canal sealer may release a soluble toxic substance to the periapical tissues and affect the local bone metabolism and wound-healing process [6]. Therefore, one of the most important prerequisites of root canal sealer is the biocompatibility, and it is important to investigate the cytotoxicity or biological irritation of the sealers. Gutta-percha has been known to have low cytotoxicity [7], however, recent studies have shown that root canal sealer containing epoxy resin and amines showed various tissue responses and cytotoxicity in vivo and in vitro [8,9]. Calcium silicate-based sealers are a sort of bioceramic based sealers that are composed of bioceramic powder and vehicle, having some mechanisms to make them biocompatible, such as biomineralization and osseo-conductivity [4].

Human periodontal ligament stem cells (hPDLSCs) are a kind of multipotent stem cell population and recently have been frequently used for in vitro biocompatibility study [10]. Meanwhile, the newly introduced injectable premixed paste type of calcium silicate-based sealers have not been investigated enough in terms of biocompatibility, although these new sealers’ physicochemical properties have been studied [11,12].

The aim of this study was to assess the effect of three calcium silicate-based sealers, EndoSeal MTA, Nano-ceramic Sealer, and Wellroot ST on various aspects, such as cell viability, inflammatory response, and osteogenic potential, of hPDLSCs. The null hypothesis was that the calcium silicate-based sealers and epoxy resin-based sealers had the similar biocompatibility. 

## 2. Materials and Methods 

### 2.1. Isolation of hPDLSCs and Cell Culture

hPDLSCs were obtained from premolars (*n* = 4) of four healthy subjects, whose ages ranged from 16 to 24 years old. Teeth were extracted under the purpose of orthodontic treatment, and the written informed consent forms according to the guidelines of the Ethics Committee of the research institution (Institutional Review Board number: 2-2017-0002) were obtained from the donors. Human PDL tissue was isolated from the middle-third of the root and minced to 2 mm size pieces using a surgical blade. Then, the pieces of hPDL tissue were washed five times using Dulbecco’s modified Eagle’s medium (DMEM; Gibco, NY, USA) containing 3% penicillin-streptomycin (Gibco) and placed on a 6-well culture plate (SPL, Pocheon, Korea). The explants were grown in DMEM with 15% fetal bovine serum (FBS; Gibco) and 1% penicillin-streptomycin. Migrated hPDLSCs from the explants were carried out to passage 2 cells by cell dissociation using trypsin (Gibco). The isolated hPDLSCs were cultured and incubated at 37 °C in 5% CO_2_. For all experiments, hPDLSCs at passages 4 to 6 were used.

### 2.2. Sample Preparation and Fresh/Setting Material Extraction Medium

Two epoxy resin-based sealers (AH-Plus; DentsplySirona, Tulsa, OK, USA; AD Seal; MetaBiomed, Cheongju, Korea) and three calcium silicate-based sealers (EndoSeal MTA; Maruchi, Wonju, Korea; Nano-ceramic Sealer; B&L Biotech, Fairfax, VA, USA; and Wellroot ST; Vericom, Chuncheon, Korea) were used for the experimental materials (Table 1).

All experimental sealers prepared according to the manufacturers’ instructions were mixed with culture medium (DMEM) at a concentration of 20 mg/mL to make material extraction medium, incubated for 24 h in 5% CO_2_ at 37 °C. The supernatant of material extraction medium was filtered using a 0.2 μm pore-size syringe filter (Corning; Merck, Darmstadt, Germany). A dilution (1/4 vol/vol) of these material extraction media was prepared with culture medium.

All sealers were mixed and manipulated depending on the manufacturer’s instructions and incubated for 48 h to allow them to be set completely in an incubator (37 °C, >95% relative humidity). Set specimens of the tested sealers were made with discs (5-mm diameter and 2-mm thick, 200 mg weight) fabricated from cylindrical Teflon molds in sterile conditions. To prepare an extraction vehicle, each set sample disc was stored and incubated in DMEM solution with 10% FBS and 1% penicillin-streptomycin for 72 h in 5% CO_2_ at 37 °C. Every 24 h the medium was changed, and then the 48-h and 72-h extraction media were collected and passed through a 0.2 µm filter (Minisart, Goettingen, Germany) (Figure 1). 

### 2.3. Cell Viability and Proliferation Assay

The 3-(4,5-dimethylthiazolyl-2-yl)-2,5-diphenyltetrazolium bromide (MTT) assay was performed to investigate the influence on the cell viability, proliferation, and the toxicity of all materials on hPDLSCs. The cells (2 × 10^4^) were seeded in a 24-well plate and exposed to the extraction media (both of the fresh and set materials) for 7 days. After each experiment period, MTT solution (final concentration, 0.5 mg/mL) was added to each well and incubated for 2 h. Subsequently, the medium containing MTT solution was removed, and 300 μL dimethyl sulfoxide (DMSO; Merck) was added to each well to dissolve the MTT formazan. The dissolved formazan solution of each well was transferred to a 96-well plate and measured at 540 nm in microplate reader (BioTek, Winooski, VT, USA).

### 2.4. Morphological Evaluation by Scanning Electronic Microscopy

All sealers were made with discs (5-mm diameter and 2-mm thick) fabricated from cylindrical Teflon molds in sterile conditions. After transferring the disc into the 96-well plate (Merck), hPDLSCs (5 × 10^4^ cells/mL) were directly seeded onto each set disc and incubated in 5% CO_2_ at 37 °C for 72 h. The discs were taken out from 96-well plate, rinsed with PBS and fixed with 2% glutaraldehyde. To examine the cell morphology on the discs using scanning electronic microscopy (SEM; Hitachi S-3000N; Hitachi, Tokyo, Japan), all discs were subjected to dehydration, air-drying, and 100-nm-thick gold/palladium coating. The magnifications used were 200× and 1000×. 

### 2.5. Confirmation of Inflammatory Response by Enzyme-Linked Immunosorbent Assay

To evaluate the inflammatory response of all materials, enzyme-linked immunosorbent assay (ELISA) was conducted. hPDLSCs (1 × 10^5^ cells) were seeded in a 6-well plate, and then after 24 h the medium was changed to 1/4 diluted fresh extraction medium and incubated for 24 h. The concentrations of IL-1β, IL-6, IL-8, IL-10, and TNF-α in the fresh extraction medium were analyzed using ELISA kit (R&D systems) according to the manufacturer’s recommendations and measured at 450 nm in a microplate reader (BioTek).

### 2.6. Measurement of Mesenchymal Phenotype by Flowcytometry Analysis

To confirm the mesenchymal phenotypic characterization of hPDLSCs, flow cytometry (FACS verse; BD bioscience) assay was carried out. The expression of mesenchymal stem cell surface molecules (CD makers as follows; CD73, CD90, CD105, CD11b, CD19, CD34, CD45, and HLA-DR) was analyzed. hPDLSCs (1 × 10^5^ cells) were seeded in a 6-well plate, and 24 h later, medium was changed to 1/4 diluted fresh extraction medium and incubated in 5% CO_2_ at 37 °C for 72 h. Then, the cells were harvested using 0.05% Trypsin-EDTA, washed with PBS, and made a single-cell suspension for CD markers antigen-antibody reaction and FACS analysis. The reaction of the CD makers in hPDLSCs was evaluated using BD Stemflow human MSC analysis kit (BD Biosciences, Piscataway, NJ, USA) and performed according to the manufacturer’s instructions. All data were analyzed using FlowJo software (BD Biosciences).

### 2.7. Quantification of Osteogenic Genes by Real Time qPCR

To find the effects of three calcium silicate-based sealers (EndoSeal MTA, Nano-ceramic Sealer, and Wellroot ST) on osteoblastic differentiation, mRNA expression of differentiation markers was evaluated. hPDLSCs were cultured for 3, 6, and 9 days in order to confirm the pattern of osteogenic genes for all sealers, with the results being analyzed using real time qPCR (RT-qPCR). hPDLSCs (1 × 10^5^ cells) were placed in a 6-well plate, and incubated for 3, 6, and 9 days in the prepared medium as follows: 1/4 diluted fresh extraction medium including with 100 µM L-ascorbic acid 2-phosphate (Merck), 10 mM b-glycerol phosphate (Sigma), and 100 nM dexamethasone (Sigma). The mRNA expression levels of alkaline phosphatase (ALP) and Runt-related transcription factor 2 (RUNX2) were determined by real time qPCR taking the β-actin encoding gene as internal reference. Isolation of mRNA and transcription into cDNA was performed using the Rneasy mini kit (Qiagen, Hilden, Germany), and revertAid first strand cDNA synthesis kit (Invitrogen). qPCR was performed on QuantStudio 3 Real-Time qPCR systems (Applied Biosystems) in paired reactions using the Taqman gene expression assays ALP, Hs00758162_m1; RUNX2, Hs01047943_m1; β-actin, Hs01060665_g1 (Thermo Fisher Scientific, Waltham, MA, USA). Changes in the expression of target genes were calculated using the 2−ΔΔ*C*_t_ method. 

### 2.8. Alkaline Phosphatase (ALP) Staining

Alkaline phosphatase (ALP) staining was conducted to measure the osteogenic activity. hPDLSCs (2.5 × 10^4^ cells) were placed in a 24-well plate and incubated for 3, 6, and 9 days in the fresh extraction medium containing osteogenic-inducing reagents (100 µM l-ascorbic acid 2-phosphate, 10 mM b-glycerol phosphate, and 100 nM dexamethasone). After each time period, cells were fixed with 4% PFA for 10 min, washed twice in PBS, and stained using the ALP Staining Kit (Merck) according to the manufacturer’s recommendations.

### 2.9. Statistical Analysis

Data were analyzed with SPSS software (version 11.0; SPSS, Chicago, IL, USA) for the comparison between materials and the test periods. After checking the data normality, statistical significance was evaluated by Friedman test and Dunn post-hoc for the comparison of materials. The differences between test periods were evaluated by Kruskal–Wallis H and post-hoc Bonferroni–Dunn test (*p* < 0.05).

## 3. Results

### 3.1. Cell Viability

The initial toxicity of all of the substances was analyzed using a fresh extraction medium, and the experiment was performed for a total of 7 days. As shown in Figure 1, AH-Plus showed the lowest cell viability through all experimental periods among all of the tested sealers. The cell viability of AH-Plus, Wellroot ST, and EndoSeal MTA were significantly decreased by time (*p* < 0.05) (Figure 2).

To evaluate the toxicity after setting of the materials, two different set extraction mediums (48- and 72-h extraction) were used. In the 48-h extraction media, hPDLSCs showed no significant differences in cell viability among all of the tested materials and control for the experimental periods (1 to 7 days) (Figure 3A). In the 72-h extraction media, however, the cell viability of Wellroot ST was the highest at 3 days, and that of Nano-ceramic Sealer was significantly increased for 7 days (*p* < 0.05) (Figure 3B). The increase in cell viability indicates the proliferation of hPDLSCs.

### 3.2. Cell Attachment and Morphology

Surface analysis on the morphology and adhesion of hPDLSCs on the fully set material disc was performed by scanning electron microscope (SEM). After incubation, hPDLSCs being attached on three calcium silicate set discs (EndoSeal MTA, Nano-ceramic Sealer, and Wellroot ST) were observed, whereas set surface of AH-Plus and AD Seal discs did not show any cell attached due to the cell death (Figure 4). Wellroot ST showed the highest level of cell adhesion and the morphology of attached cells with well-spread and flattened structure.

### 3.3. Inflammatory Response

IL-1β, IL-10, and TNF-α were not expressed in all experimental groups among pro-inflammatory cytokines (IL-1β, IL-6, IL-8, IL-10, and TNF-α). AH-Plus presented a significantly higher expression of IL-6 and IL-8 than the other sealers (*p* < 0.05) (Figure 5).

### 3.4. Mesenchymal Phenotype Expression

hPDLSCs were cultured for 3 days in different types of endodontic sealer eluates (1/4 diluted fresh material extraction medium). Fresh extraction medium of the AH-plus showed high toxicity in hPDLSCs (Figure 2), therefore, it was excluded from mesenchymal phenotypic analysis. hPDLSCs cultured in fresh extract media (AD Seal, EndoSeal MTA, Nano-ceramic Sealer, and Wellroot ST) were observed similar to normal hPDLSCs used as a positive control (red peak). Their phenotypic characterization showed that the mesenchymal stem cell (MSC) markers, CD105, CD73, and CD90 were high expressed, while the hematopoietic markers, CD45, CD34, CD14, CD19 and HLA-DR were low expressed (Figure 6).

### 3.5. Osteogenic Potential

The mRNA expression of RUNX2 did not show any significant difference among the materials (EndoSeal MTA, Nano-ceramic Sealer, and Wellroot ST) and time periods (3, 6, and 9 days) (Figure 7A). However, ALP mRNA expression showed a significant increase in time-dependent manner on all of three calcium silicate-based sealers (*p* < 0.05) (Figure 7B). In addition, ALP staining showed increasing time-dependent deep-purple coloration in the three calcium silicate-based sealers (Figure 8). 

## 4. Discussion

Contemporary root canal treatment has been advanced not only for the clinical convenience but also for good prognosis from the development of various materials and techniques. Among the newly introduced materials, calcium silicate-based sealers are the most recently introduced representative materials, but there is still a lack of evidence of their characteristics and clinical use. From this study, calcium silicate-based sealers, such as EndoSeal MTA, Nano-ceramic Sealer, and Wellroot ST, appeared to be more biocompatible and less cytotoxic than epoxy resin-based sealers.

The calcium silicate-based sealer is usually supplied in a single paste syringe, which clinicians use directly to the root canal by injection [3]. Thus, this type of sealer may extrude to the periapical tissue, and sealer puff could occur when it is applied in a clinical situation. This means that the sealer may contact cells in PDL space directly due to its flowability, which may cause an acute inflammatory reaction on periapical area or spread to periapical bony lesions [2]. Therefore, this study was performed to compare the newly introduced calcium silicate-based root canal sealers to the conventional epoxy resin-based sealers for their biocompatibilities to hPDLSCs.

Cell viability is defined as a percentage of live cells in a whole population and usually used to evaluate the toxicity of materials [13]. Various setting time is needed for a complete set of each root canal sealer, which may affect cell viability according to their different setting procedures [4,14]. In this study, these epoxy resin-based sealers were used as the control group for three calcium silicate sealers. AH-Plus is one of the epoxy resin-based endodontic sealers that is known to have lower biocompatibility than the calcium silicate-based sealers [15,16], whereas AD Seal has been demonstrated to have lower inflammatory response than AH-Plus [17,18]. Fresh extraction mediums of all tested sealers were used to confirm the initial toxicity of the materials, while two different extraction (48- and 72-h) mediums of set sealers were used to identify the toxicity after setting over time. 

Initial toxicity of epoxy resin-based sealers immediately after mixing and before complete set is well known because of the release of the sealer’s amine or epoxy resin components [2,17]. In the present study, AH-Plus showed the lowest cell viability on hPDLSCs in all time periods in fresh media than the other four tested sealers, whereas the cell viability in AD Seal was not significantly different from the control. AH-Plus might have shown a highly cytotoxic effect before complete set because epoxy resin, one of the major components of these materials, acts as a toxic factor [2]. However, AD Seal containing calcium phosphate showed it would be more biocompatible than the other epoxy resin sealers, such as AH-Plus [17,19]. Calcium phosphate was reported as a highly biocompatible material having a bone-inducing effect, which could be less harmful when contacted with apical tissue [20]. 

In the present study, even calcium silicate-based sealers (Wellroot ST and EndoSeal MTA) showed decreased cell viability by time in fresh media, which might be a result of their high pH in the fresh state. Calcium silicate-based sealers have a high alkalinity because they dissolve into calcium hydroxide when in contact with soft tissues [21]. Although the high pH of root canal sealers might have this negative effect on cell viability, it may provide several biological advantages. The high alkalinity of these sealers may change the environment to promote hard tissue formation and to interfere with osteoclastic activity in the adjacent tissues, which favors healing [21].

Considering all results from this study, the initial contact of periodontal tissue with the fresh mixed sealers would be worse than that with set sealers. In general, clinical situations, the contact area with the sealer is limited around the apical region [2], which may not be harmful to the survival and proliferation of hPDLSCs in the apical periodontal tissues, especially after the sealer has been fully set. However, AH-Plus could have a detrimental effect on the survival of hPDLSCs in some clinical situations where the apical apex is wide or excessive sealer is expected to be exposed to the apical periodontal tissue.

The cell viabilities according to the extraction time (48-h and 72-h) from set sealers were also investigated, and it was found that there were no significant differences among all of the tested sealers with increasing extraction time. The cell viability in the 48-h extraction media was not statistically significant by all experimental periods. Even AH-Plus, which was the most toxic in fresh media state, showed decreased toxicity to the same level of the control. This indicates that cytotoxicity of AH-Plus would be decreased after being fully set. In 72-h extracted media after set, Wellroot ST showed significantly higher cell viability than control at 3 days, and the viable cells were significantly increased for 7 days in Nano-ceramic sealer. Therefore, it may be implicated that calcium silicate-based sealers have a good effect on the survival and proliferation of hPDLSCs. However, further studies are needed to clarify intracellular signaling pathways involved in this cell proliferation by calcium silicate sealers.

SEM observations in this study showed the death of all hPDLSCs on the surface of AH-Plus and AD Seal set specimen. These findings indicated that two epoxy resin-based sealers have toxicity on hPDLSCs even after having been fully set. On the other hand, the cells were well attached and proliferated on the set surface of three calcium silicate-based sealers, which means that they would be more advantageous to cell biocompatibility even in direct contact conditions than epoxy resin-based sealers. Wellroot ST was found to be the most effective for attachment of hPDLSCs on the set surface. The favorable cell attachment and proliferation on Wellroot ST and Nano-ceramic Sealer might be due to their smooth surface. However, they might have been difficult to be attached and differentiated on EndoSeal MTA, which has too rough a surface. In addition, the ability of cells to attach to the surface of the sealer appears to be influenced by the chemical composition of the sealer [22].

Pro-inflammatory cytokines are involved in the up-regulation of inflammatory reactions. There is abundant evidence that certain pro-inflammatory cytokines, such as IL-1β, IL-6, and TNF-α, are involved in the process of pathological pain [23]. It is known that inflammatory processes are initiated and maintained by pro-inflammatory cytokines which have distinct or shared biological activities [24]. In the present study, two pro-inflammatory cytokines, IL-6 and IL-8, from hPDLSCs exposed to AH-Plus were secreted significantly higher than the other sealers, which is markedly associated with low cell viability in AH-Plus. These results were consistent with the cell viability in fresh media demonstrating the highest cytotoxicity of AH-Plus. Except for AH-Plus, the other sealers (AD Seal, EndoSeal MTA, Nano-ceramic Sealer, and Wellroot ST) secreted IL-6 and IL-8 as much as the control group. Among three calcium silicate-based sealers, EndoSeal MTA showed lower levels of IL-6 and IL-8 secretion compared to the other two sealers. 

Sealers with the ability to enhance osteogenesis have the potential to promote faster and more predictable healing of apical periodontitis. Thus, the evaluation of sealer for these potentials can improve a clinician’s ability to choose an endodontic sealer that will not only minimally inhibit wound healing but also possibly help expedite the process.

Mesenchymal stem cells (MSC) are known to be isolated from bone marrow, peripheral blood, placenta, and dental pulp, etc. [25]. The hPDLSCs are a kind of multipotent stem cell and recently have been frequently used in dentistry not only for in vitro study but also for regenerative procedures and/or trials. The adult hPDLSCs have been demonstrated to express mesenchymal surface markers and have multipotent capacity to differentiate into adipocyte, osteoblast-like, and cementoblast-like cells, which form cementum/PDL like tissues [10]. In the standard culture condition, hPDLSCs have high expression of MSC markers, such as CD105, CD73, and CD90, but low expression of hematopoietic stem cell makers such as CD45, CD34, CD14, CD19, and HLA class II [10,22,25]. Three calcium silicate-based sealers tested in the present study did not interfere with these behaviors of hPDLSCs that differentiate into mesenchymal phenotypes. In addition, these results are consistent with the previous report [26].

It is desirable for root canal sealers to present the capacity of modulating the periapical environment [26]. RUNX2 is a key transcription factor associated with osteoblast differentiation. It regulates the osteogenic genes, related with the initiation and termination of osteoblast cell cycle, which plays a role in regulating the cell proliferation [27]. This study investigated the osteogenic molecule presented during the hPDLSCs differentiation to osteoblasts. There was no significant difference of RUNX2 expression among experimental periods and also among sealers. We can conclude that the three calcium silicate sealers did not interfere with the expression level of RUNX2 for hPDLSCs.

ALP is a factor for evaluating an initial cell differentiation into osteoblasts [3]. RT-qPCR showed that all of the tested calcium silicate sealers recovered the ALP mRNA expression level of hPDLSCs in the differentiation media after 6 days. That of EndoSeal MTA was lower than the other two calcium silicate sealers at 3 days, but differences of the level were not significant at 9 days. In order to visualize the ALP activity, ALP staining was performed on a representative cell. The data were consistent with the results of RT-qPCR, which indicates that the three calcium silicate-based sealers have osteoblastic potential in differentiation media with the elevated ALP levels in the experimental periods. 

Considering above the results, the three calcium silicate sealers do not seem to interfere with the differentiation of hPDLSCs into osteoblasts and the expression of osteoblastic markers present in the apical tissues, enabling favorable healing of periapical lesion. In particular, Nano-ceramic Sealer has favorable initial osteoblastic potential, which is more beneficial for initial periapical healing than other sealers.

Thanks to the advantageous features of the calcium silicate-based sealers, a new canal-filling technique using single cone with or without compaction has been suggested as an up-to-date alternative to the current standard of warm condensation techniques. Future in vitro or clinical studies would be necessary to strengthen the rationale for the new materials and techniques. 

## 5. Conclusions

Based on the results from this study, calcium silicate-based sealers (EndoSeal MTA, Nano-ceramic Sealer, Wellroot ST) appear to be more biocompatible and less cytotoxic than epoxy resin-based sealers. 

## Figures and Tables

**Figure 1 materials-12-02411-f001:**
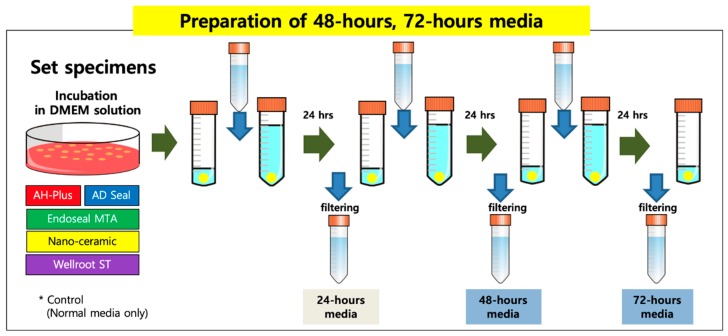
Preparation of set material extraction media. Each set sample disc was stored and incubated in Dulbecco’s modified Eagle’s medium (DMEM) solution. Every 24 h the medium was changed, and then the 48-h and 72-h extraction media were collected.

**Figure 2 materials-12-02411-f002:**
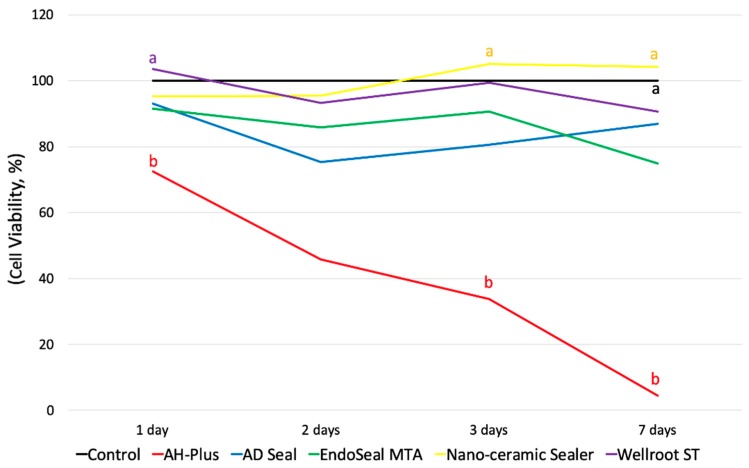
Cell viability of human periodontal ligament stem cells (hPDLSCs) by 3-(4,5-dimethylthiazolyl-2-yl)-2,5-diphenyltetrazolium bromide (MTT) assay using extraction media derived from the fresh sealers for 7 days (**a**, **b**; different alphabet indicates the significant difference between the sealers. *p* < 0.05). AH-Plus showed the lowest cell viability through all experimental periods among all of the tested sealers). Wellroot ST on 7 days shows significantly lower viability than 1 day. EndoSeal MTA on 7 days shows significantly lower viability than 3 days.

**Figure 3 materials-12-02411-f003:**
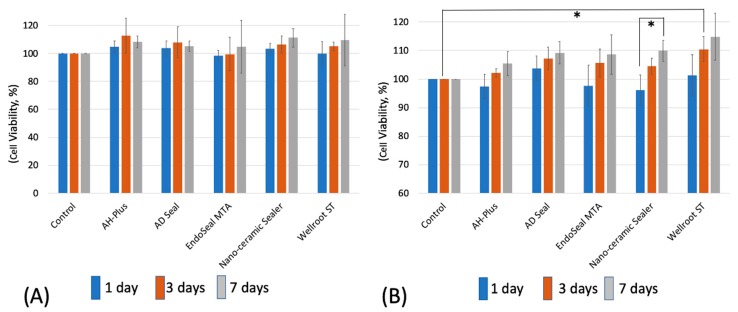
Cell viability of hPDLSCs by MTT assay using extraction media derived from the set sealers at (**A**) 48 h and (**B**) 72 h for 7 days (* means significant difference between groups connected by bar. *p* < 0.05).

**Figure 4 materials-12-02411-f004:**
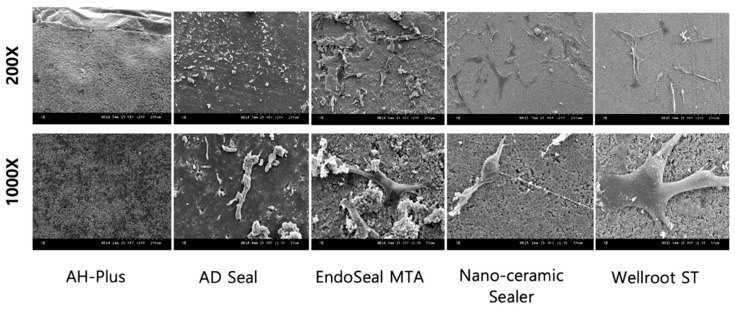
The morphology of hPDLSCs attached on the surface of five root canal sealers after culture for 3 days at a magnification of 200× (upper row) and 1000× (lower row). Cell death was observed (no cell adhesion detected) in AH-Plus and AD Seal, while three calcium silicate-based sealers allowed well-adhered cells with high degree of cell spreading and production of extracellular matrix.

**Figure 5 materials-12-02411-f005:**
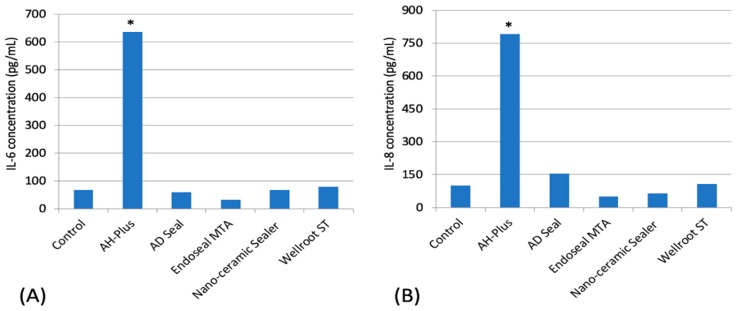
The effects of sealers on the expression of pro-inflammatory cytokines (**A**) IL-6 and (**B**) IL-8 in hPDLSCs (* means significant difference between groups connected by bar. *p* < 0.05).

**Figure 6 materials-12-02411-f006:**
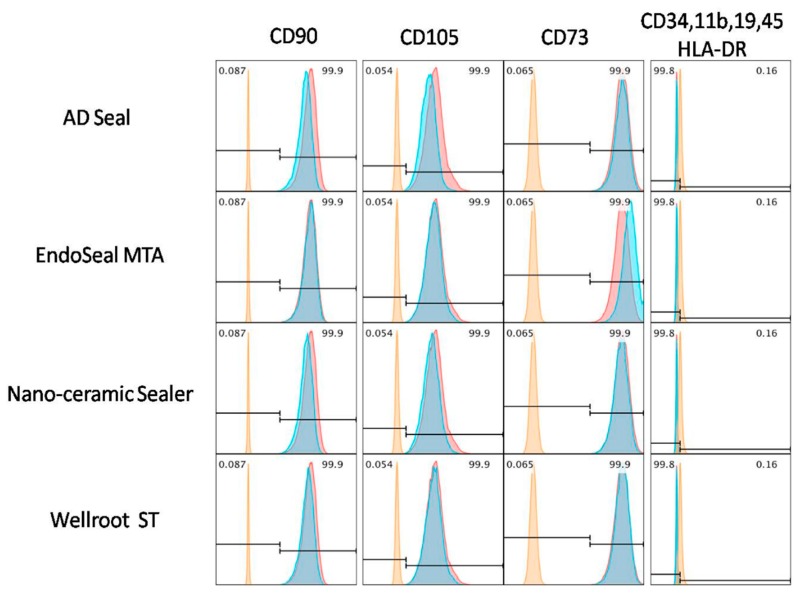
Mesenchymal phenotype analysis of hPDLSCs using flowcytometry. After culture in 1/4 diluted extraction medium from fresh sealers with four root canal sealers for 3 days, hPDLSCs showed phenotypic characterization to have high expression levels of the mesenchymal stem cell markers CD90, CD105, and CD73, but low expression levels of the hematopoietic markers CD34, CD11b, CD19, CD45, and HLA-DR.

**Figure 7 materials-12-02411-f007:**
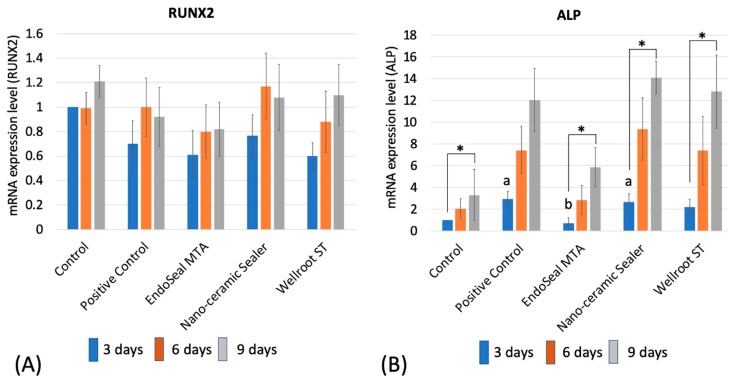
The effects of sealers on the mRNA expression of (**A**) RUNX2 and (**B**) alkaline phosphatase (ALP) by real time qPCR for 9 days (**a**, **b** mean significant difference between groups connected by bar. * means significant difference between time periods. *p* < 0.05). Three calcium silicate-based sealers showed no significant difference with control and positive control on RUNX2 expression. All of the Nano-ceramic Sealer, EndoSeal MTA, and Wellroot ST showed significantly higher ALP expression on 9 days than on 3 days.

**Figure 8 materials-12-02411-f008:**
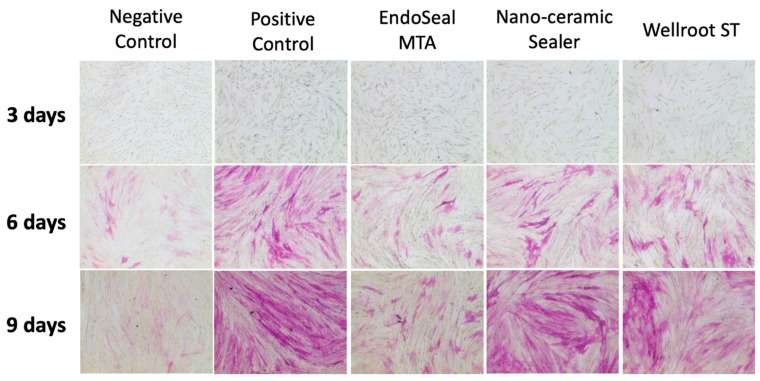
The effects of sealers on osteoblastic differentiation in a representative cell of hPDLSCs for 9 days using ALP staining. Wellroot ST and Nano-ceramic Sealer showed staining strength similar to that of positive control on 9 days.

**Table 1 materials-12-02411-t001:** Tested sealers and their composition in this study.

Sealer	Manufacturer	Composition	
AH-Plus	DentsplySirona, Tulsa, OK, USA	Paste A 25–50% bisphenol A 10–25% zirconium dioxide NS calcium tungstate NS iron oxide	Paste B 2.5–10% N, n-dibenzyl-5-oxanonandiamin-1,9 2.5–10% amantadine
AD Seal	MetaBiomed, Cheongju, Korea	Base <20% epoxy resin NS calcium phosphate NS zirconium dioxide NS calcium oxide NS ethylene glycol salicylate	Catalyst 2.5–10% N, n-dibenzyl-5-oxanonandiamin-1,9 2.5–10% amantadine
EndoSeal MTA	Maruchi, Wonju, Korea	Calcium silicates, calcium aluminates, calcium aluminoferrite, calcium sulfates, radiopacifier, thickening agents	
Nano-ceramic Sealer	B&L Biotech, Fairfax, VA, USA	Calcium silicates, zirconium oxide, filler, thickening agent.	
Wellroot ST	Vericom, Chuncheon, Korea	Calcium silicate compound, calcium sulfate dehydrate, calcium sodium phosphosilicate, zirconium oxide, titanium oxide, thickening agents	

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
