# Peer review of "In Vitro Comparison of Biocompatibility of Calcium Silicate-Based Root Canal Sealers"

_materials, 2019, doi:10.3390/ma12152411_

Reviewer 1 Report

Thank you for the submission of the present interesting manuscript. The manuscript is well written and all the sections of the manuscript are presented adequately. 

I would like however to comment on Figure 4 which I believe there is a point that needs further clarification. The fact that we do not see attached cells on the surface of AH plus means that the cells are dead? Please clarify that point  

Author Response

Thank you for the submission of the present interesting manuscript. The manuscript is well written and all the sections of the manuscript are presented adequately. 

I would like however to comment on Figure 4 which I believe there is a point that needs further clarification. The fact that we do not see attached cells on the surface of AH plus means that the cells are dead? Please clarify that point.

 Authors’ responses: Thank you for this kind comments. We have revised the sentence about the reason in the manuscript and figure legend (your idea is correct, all the cells were dead). 

Reviewer 2 Report

In general, the idea and innovation of this study, regards the analysis of biocompatibility of calcium silicate-based root canal sealers is interesting, because the management of this materialconditions is validated but further studies on this topic could be an innovative issue in this field could be open an innovative matter of debate in literature by adding new information. Moreover, there are few reports in the literature that studied this interesting topic with this kind of study design.

The study was well conducted by the authors; However, there are some concerns to revise that are described below.

The introduction section resumes the existing knowledge regarding the important factor linked with periodontal ligament stem cells.

However, as the importance of the topic, the reviewer strongly recommends, before a further re-evaluation of the manuscript, to update the literature through read, discuss and cites in the references with great attention all of those recent interesting articles, that helps the authors to better introduce and discuss the aim of the study in light of the importance of microscopy and histology of biomaterials in dental field: 1) Lo Giudice G, Lo Giudice R, Matarese G, Isola G, Cicciù M, Terranova A, Palaia G, Romeo U. Evaluation of magnification systems in restorative dentistry. An in-vitro study. Dental Cadmos 2015, 83:296-305. 2) Kadić S, Baraba A, Miletić I, Ionescu A, Brambilla E, Ivanišević Malčić A, Gabrić D. Push-out bond strength of three different calcium silicate-based root-end filling materials after ultrasonic retrograde cavity preparation. Clin Oral Investig. 2018 Apr;22(3):1559-1565. doi: 10.1007/s00784-017-2244-6. 3) Ramaglia L, Saviano R, Matarese G, Cassandro F, Williams RC, Isola G.Histologic Evaluation of Soft and Hard Tissue Healing Following Alveolar Ridge Preservation with Deproteinized Bovine Bone Mineral Covered with Xenogenic Collagen Matrix. Int J Periodontics Restorative Dent. 2018 September/October;38(5):737–745. doi: 10.11607/prd.3565.

The authors should be better specified, at the end of the introduction section, the rational of the study and the aim of the study with the null hypothesis. In the material and methods section, should better clarify how was performed the setting material extraction medium and the scanning electronic microscopy. Moreover, specify if data were normalized or not. Please specify if was used a test unit.

The discussion section appears well organized with the relevant paper that support the conclusions, even if the authors should better discuss the importance of stem cells in dentistry. The conclusion should reinforce in light of the discussions.

In conclusion, I am sure that the authors are fine clinicians who achieve very nice results with their adopted protocol. However, this study, in my view, does not in its current form, satisfy a very high scientific requirement for publication in this journal and requests a revision before publication.

 Minor Comments:

 Abstract:

-          Better formulate the introduction section by better describing the background

 Introduction:

-          Please refer to major comments

Discussion

-          Please add a specific sentence that clarifies the results obtained in the first part of the discussion

-          Page 11 last paragraph: Please reorganize this paragraph that is not clear

Author Response

In general, the idea and innovation of this study, regards the analysis of biocompatibility of calcium silicate-based root canal sealers is interesting, because the management of this materialconditions is validated but further studies on this topic could be an innovative issue in this field could be open an innovative matter of debate in literature by adding new information. Moreover, there are few reports in the literature that studied this interesting topic with this kind of study design.

The study was well conducted by the authors; However, there are some concerns to revise that are described below.

The introduction section resumes the existing knowledge regarding the important factor linked with periodontal ligament stem cells.

However, as the importance of the topic, the reviewer strongly recommends, before a further re-evaluation of the manuscript, to update the literature through read, discuss and cites in the references with great attention all of those recent interesting articles, that helps the authors to better introduce and discuss the aim of the study in light of the importance of microscopy and histology of biomaterials in dental field: 1) Lo Giudice G, Lo Giudice R, Matarese G, Isola G, Cicciù M, Terranova A, Palaia G, Romeo U. Evaluation of magnification systems in restorative dentistry. An in-vitro study. Dental Cadmos 2015, 83:296-305. 2) Kadić S, Baraba A, Miletić I, Ionescu A, Brambilla E, Ivanišević Malčić A, Gabrić D. Push-out bond strength of three different calcium silicate-based root-end filling materials after ultrasonic retrograde cavity preparation. Clin Oral Investig. 2018 Apr;22(3):1559-1565. doi: 10.1007/s00784-017-2244-6. 3) Ramaglia L, Saviano R, Matarese G, Cassandro F, Williams RC, Isola G.Histologic Evaluation of Soft and Hard Tissue Healing Following Alveolar Ridge Preservation with Deproteinized Bovine Bone Mineral Covered with Xenogenic Collagen Matrix. Int J Periodontics Restorative Dent. 2018 September/October;38(5):737–745. doi: 10.11607/prd.3565.

Authors’ responses: Thank you for this kind suggestions with detailed explanation. We have revised the introduction section based on the suggested reference. We have searched the article of “Dental Cadmos 2015, 83:296-305” and found it is written in Italian so that unfortunately we cannot cite it in this manuscript. And the article “Int. J. Periodontics Restorative Dent. 2018, 38, 737–745” is from the periodontology and dealing with the collagen matrix, so that we just cited the one from the Clin Oral Investig;

Kadić S.; Baraba A.; Miletić I.; Ionescu A.; Brambilla E.; Ivanišević Malčić A.; Gabrić D. Push-out bond strength of three different calcium silicate-based root-end filling materials after ultrasonic retrograde cavity preparation. Clin. Oral. Investig. 2018, 22, 1559–1565. 

 The authors should be better specified, at the end of the introduction section, the rational of the study and the aim of the study with the null hypothesis. In the material and methods section, should better clarify how was performed the setting material extraction medium and the scanning electronic microscopy. Moreover, specify if data were normalized or not. Please specify if was used a test unit.

The discussion section appears well organized with the relevant paper that support the conclusions, even if the authors should better discuss the importance of stem cells in dentistry. The conclusion should reinforce in light of the discussions.

In conclusion, I am sure that the authors are fine clinicians who achieve very nice results with their adopted protocol. However, this study, in my view, does not in its current form, satisfy a very high scientific requirement for publication in this journal and requests a revision before publication.

 Authors’ responses: Thank you for these comments for each section of introduction, M&M, discussion and conclusion. We have revised considering the points of critics. 

  Minor Comments:

 Abstract:

-          Better formulate the introduction section by better describing the background 

Introduction:

-          Please refer to major comments

Discussion

-          Please add a specific sentence that clarifies the results obtained in the first part of the discussion

-          Page 11 last paragraph: Please reorganize this paragraph that is not clear

 Authors’ responses: Thank you for these comments. We have revised accordingly. 

Reviewer 3 Report

Dear author,

Congratulations for the research that is interesting and clinically relevant, but the manuscript’s need to be revised.

Line 88 need to be revised: The sealers were mixed with culture medium, after preparation according the manufactures’ instructions?

Line 95: What was the temperature and humidity used?

Line 98: Why you did not sterilize the pelets?

Line 333: Correct the sentence.

Author Response

Congratulations for the research that is interesting and clinically relevant, but the manuscript’s need to be revised.

 Line 88 need to be revised: The sealers were mixed with culture medium, after preparation according the manufactures’ instructions?

Authors’ responses: Thank you for this notice. We have detailed . 

Line 95: What was the temperature and humidity used?

Authors’ responses: We have added the information.

Line 98: Why you did not sterilize the pelets?

Authors’ responses: We have actually sterilized all the related materials. 

Line 333: Correct the sentence.

Authors’ responses: Thank you for this notice. 

Round  2

Reviewer 2 Report

In the R1 version of the manuscript entitled: “Comparison of Biocompatibility of Calcium Silicate-Based Root Canal Sealers” the authors followed all the issues suggested by the reviewer. Though the changes based on the reviewer comments, almost of the criticisms were carefully analysed and solved.

I have carefully evaluated all parts of the manuscript. I believe that the article, in this version, is now adequate for publication in this journal.